# MAPK Signaling Pathway in Oral Squamous Cell Carcinoma: Biological Function and Targeted Therapy

**DOI:** 10.3390/cancers14194625

**Published:** 2022-09-23

**Authors:** Yuxi Cheng, Juan Chen, Yuxin Shi, Xiaodan Fang, Zhangui Tang

**Affiliations:** 1Xiangya Stomatological Hospital, Central South University, Changsha 410008, China; 2Xiangya School of Stomatology, Central South University, Changsha 410008, China

**Keywords:** oral squamous cell carcinoma, MAPK, signaling pathway, immunotherapy

## Abstract

**Simple Summary:**

The aim of the present review was to summarize our studies on the signaling pathways in cancer types that are involved in the development of oral cancer and several mitogen-activated protein kinase-related molecular targeting technologies combined with immune checkpoint therapies to provide new therapeutic strategies for oral squamous cell carcinoma. In addition, we have provided a reasonable outlook, with a systematic basis, for future diagnosis and accurate treatment.

**Abstract:**

Oral squamous cell carcinoma accounts for 95% of human head and neck squamous cell carcinoma cases. It is highly malignant and aggressive, with a poor prognosis and a 5-year survival rate of <50%. In recent years, basic and clinical studies have been performed on the role of the mitogen-activated protein kinase (MAPK) signaling pathway in oral cancer. The MAPK signaling pathway is activated in over 50% of human oral cancer cases. Herein, we review research progress on the MAPK signaling pathway and its potential therapeutic mechanisms and discuss its molecular targeting to explore its potential as a therapeutic strategy for oral squamous cell carcinoma.

## 1. Introduction

Oral squamous cell carcinoma (OSCC) accounts for 95% of cases of head and neck squamous cell carcinoma (HNSCC). It is highly malignant and aggressive, with a poor prognosis and a 5-year survival rate under 50% [1,2]. Despite advances in its diagnosis and treatment, the overall prognosis or survival of patients has not improved [3,4]. Currently, the treatment of OSCC is mainly surgery, combined with radiotherapy and chemotherapy. The treatment method is affected by many factors, particularly the balance between the surgical effect and adverse reactions [5,6]. Therefore, to improve the diagnostic efficiency and obtain a better prognosis, a comprehensive study should be conducted on the molecular mechanism of OSCC, and novel diagnostic tools and accurate treatment methods should be explored.

Protein kinases are enzymes related to protein degradation. Their enzymatic activity is controlled by phosphorylation, which can covalently connect phosphates to the side chains of serine, threonine, or tyrosine of specific proteins in cells [7]. The basic composition of the mitogen-activated protein kinase (MAPK) pathway is divided into three modules in sequence, with a cascade effect: MAPK kinase kinase (MAPKKK), MAPK kinase (MAPKK), and MAPK [8]. Currently, 25 MAPKKKs, seven MAPKKs, and 13 MAPKs have been identified. In multicellular organisms, the three classical subfamilies of MAPKs have been described in detail. In recent years, novel MAPK members, such as extracellular signal-regulated kinases (ERKs) 5 and 7, have been elucidated. They play important roles in controlling many physiological processes. Currently, six groups of members have been identified, i.e., ERK1/2, ERK3/4, ERK5, ERK7/8, Jun N-terminal kinase (JNK) 1/2/3, and p38 α/β/γ (ERK6)/δ [9,10]. They are involved in regulating cell activity by phosphorylating specific serine and threonine residues found on target protein substrates, including gene expression, mitosis, exercise, metabolism, and programmed death [11].

Herein, we review cellular and molecular mechanisms involved in activating and regulating the MAPK signaling pathway in OSCC and related drug targets, its role in tumor progression, and the related molecular targeted therapy.

## 2. MAPK Signaling Pathway

### 2.1. Activation

MAPK activation requires double phosphorylation on the Thr-X-Tyr (X representing any amino acid) motif catalyzed by MAP2K. Upon activation, MAPK phosphorylates specific serine and threonine residues on target substrates, including other protein kinases and many transcription factors. Common and bispecific phosphatases inactivate MAPKs and are regulated by scaffold proteins [12]. Figure 1 demonstrates the cascade activation of the MAPK signaling pathway.

Currently, the ERK1/2 pathway is the most widely studied pathway in the MAPK family. It involves extracellular growth factors, such as the epidermal growth factor (EGF), and activated tyrosine kinase receptors, such as the EGF receptor (EGFR), providing a binding site for the adaptor protein growth factor receptor-bound protein 2 and recruiting the SOS protein into the cell membrane. SOS activates Ras by consuming GTP to form Ras GTP. After Ras activation, it can affect many downstream proteins, including AF6, phosphatidylinositol-3-kinase (PI3K), phospholipase C, and Raf. Among them, Ras GTP recruits the Raf protein into the plasma membrane and phosphorylates it using other kinases (protein kinase A, p21-activated kinase, Src) to activate its kinase function. The Raf protein family includes B-Raf, A-Raf, and C-Raf (Raf1). B-Raf plays an important role in the development and progression of malignant tumors. The activated Raf kinase binds to the downstream MEK 1/2 and activates ERK1/2. Activated ERK1/2 can continue to phosphorylate transcription factors, such as ELK1, ETS, FOS, Jun, myc, and Sp1, and induce gene expression related to cell cycle and cell proliferation. In addition, activated ERK1/2 can phosphorylate various intracellular kinases, such as Raks, msks, and mnks, which affect cell proliferation and adhesion [13,14,15].

The JNK/MAPK signaling pathway can activate cytokines (tumor necrosis factor-alpha [TNF-α], interleukin [IL]-1), EGF, and some G protein-coupled receptors by generating stress through, for example, ultraviolet light, heat shock, hyperosmotic stimuli, and protein synthesis inhibitors. The stress response signal is transmitted to MAPKKK through the Rho subfamily (Rac, rho, Cdc42), a member of the small molecule G protein Ras superfamily, which in turn activates mek4/7 and JNK. JNK phosphorylation can act on various downstream transcription factors (e.g., Jun, ELK1, Ets2, etc.) and kinases (mainly MNK) and produce various physiological processes that promote cell growth, differentiation, survival, and apoptosis [16].

The main inducing factors of the p38/MAPK pathway are hypoxia, ultraviolet radiation, osmotic shock, inflammation, and other stress reactions. P38 MAPK is mainly activated by ERK3/6 phosphorylation, which further promotes cell apoptosis and inhibits cell proliferation by inducing transcription factors and kinases. In addition, this pathway also promotes cell movement [17]. Ras plays an important role in activating the MAPK signal pathway. It is a key component of many cellular signal transduction pathways. Its active state affects cell growth and differentiation. Its dysregulation leads to abnormal cell behavior, including increased cell growth, proliferation, dedifferentiation, and survival, thereby promoting the occurrence of cancer. Permanent activation of the Ras protein caused by mutation is prevalent in all human cancers. Therefore, Ras inhibitors are effective drugs to treat OSCC [18].

### 2.2. Mutation of the MAPK Signaling Pathway

As a signal medium in cells, MAPK controls cell differentiation, proliferation, apoptosis, and other effector functions under the stimulation of external pressure or ligands. The mutation rate of MAPK1 is higher in Asian populations, with an average of 0.79% among 32 cancer genomic profiles from The Cancer Genome Atlas (TCGA) database. The mutation rate of MAPK1 is relatively higher in HNSCC than in pan-cancer TCGA [19,20]. One-fifth of the patients with HNSCC are affected by MAPK pathway mutations, and abnormalities in the MAPK pathway are correlated with the survival time of patients [21]. Chan et al. analyzed the gene-chip technology and showed that *MAPK* in human HNSCC was overexpressed compared to matched non-cancerous tissues. The expression of several genes in MAPK signaling pathways, such as p38β, ERK2, and JNK2, increased two-folds, showing statistical significance. p38β, JNK2, and ERK2 showed a 5.27-, 2.57-, and 3.30-time increase, respectively [22]. An immunohistochemical analysis of 100s of HNSCC tissues showed elevated active phosphorylated p38 in 79% of the tissues, with increased phosphorylation activities of ERK1/2 and JNK in <33% and <16% of cases, respectively [21,23].

### 2.3. MAPK Signaling Pathway in OSCC

#### 2.3.1. ERK/MAPK Signaling Pathway in OSCC

Activation of ERK1/2 is mostly associated with cell survival, while that of JNK or p38 is associated with the induction of apoptosis [24]. However, this classification is too simplistic, and the actual role of each MAPK cascade depends highly on the cell type and the situation [25]. Activation of MAPK, particularly ERK, is differentially regulated according to the stage of tumor differentiation. For instance, the phosphorylation level of ERK is lower in advanced poorly differentiated prostate cancer than in early prostate cancer [26]. Dickkopf-related protein 3 (Dkk3), as a member of the Dickkopf WNT signaling pathway inhibitor family, has a tumor suppressive effect [27,28]. For example, overexpression of Dkk3 messenger (mRNA) is related to a good prognosis of prostatic cancer [29]. It may induce cancer cell apoptosis by overexpressing through adenovirus-mediated gene transfer [30,31]. Dkk3 plays a carcinogenic role in OSCC. Its overexpression in OSCC cells would largely increase the malignancy of the cells in vitro and in vivo and regulate the malignant behavior of cancer cells through the PI3K/mammalian target of rapamycin (mTOR)/Akt and MAPK pathways. Thus, MAPK may have a tumor- or tissue-specific effects [32]. C-Myc can alter the biological behavior of tumors through the ERK/MAPK pathway. For example, Marconi et al. showed that KRAS mutations activate the Raf/MEK/ERK signaling pathway to upregulate c-Myc, causing overexpressions of Bcl-2, hypoxia-inducible factor (HIF)-1α, vascular endothelial growth factor (VEGF), MMP-9, and other proteins, affecting the invasive, hypoxic, angiogenic, migratory, and inflammatory processes in OSCC [33].

ERK1 and ERK2 are widely expressed in tissues and participate in regulating meiosis, mitosis, and post-mitotic function of differentiated cells. In the early 1980s, ERK1 was the first MAPK core molecule identified in mammals [34]. ERK activation causes phosphorylation and activation of various cytoplasmic substrates, such as cytoskeletal proteins and downstream protein kinases. In addition, phosphorylated ERK1/2 can be transported to the nucleus to activate various transcription factors, such as ELK-1, SP-1, and AP-1, thereby regulating the transcription of different genes [10]. The ERK1/2 signaling pathway mainly affects tumor cells by proliferating cell cycle regulation. Sustained ERK activation can induce cell cycle inhibition and pro-differentiation signals in epithelial origin cells [35]. G1/S conversion is the key regulatory point of the cell cycle. Sustained activation and nuclear localization of ERK1/2 may affect G1/S conversion by regulating cyclin D1 transcription [36,37]. Inhibiting the ERK/MAPK signaling pathway leads to the proliferation, invasion, and migration of OSCC cells, causing G0/G1 arrest and promoting apoptosis [38,39]. Wu et al. [40]. reported that downregulations of MAPK/ERK1/2 and PI3K/Akt signals and cyclin D1 and E expression levels can induce G0/G1 arrest and inhibit OSCC cell proliferation. ERK1/2 is closely associated with tumor invasion and migration, and phosphorylation of ERK/MAPK activates AP-1 and nuclear factor kappa B (NF-κB). They upregulated expressions of MMP2 and MMP9, which are extracellular membrane-degrading enzymes associated with tumor aggressiveness. MMP2 and MMP9 degrade type IV collagen, a major extracellular membrane component of the basement membrane, which may be critical for tumor invasion and metastatic potential, thereby degrading the extracellular matrix and allowing cells to cross the basement membrane, facilitating tumor cell metastasis [41,42]. Blocking ERK/MAPK activation inhibited cell migration and stem characteristics of the nhri-hn1 cell line in a mouse tongue cancer model [43]. Junhai et al. found that miR-145 could inactivate the ERK/MAPK signaling pathway by inhibiting Hoxa1, thereby inhibiting the proliferation, migration, and invasion of OSCC cells and inhibiting their growth in vivo [44]. The integrin (ITG) family of proteins plays important roles in OSCC αV invasion, migration, and apoptosis via ITG-β. They regulate the proliferation and invasion of OSCC cells through the MAPK/ERK signaling pathway [45,46,47]. Chloride intracellular channel 1 (CLIC1) silence reduces αv and β1. p-ERK, vimentin, MMP2, and MMP9 levels increased p-p38, E-cadherin, Caspase3, and caspase9 levels. CLIC1 interacted with ITG, thereby activating the MAPK signaling pathway, which regulates OSCC progression [48]. Different MAPK activation times may lead to different results, occasionally even contradictory ones. For example, transient activation of ERK may generate proliferative signals, but sustained phosphorylation may generate signals leading to cell differentiation [49].

#### 2.3.2. JNK/MAPK Signaling Pathway in OSCC

JNKs were isolated and identified as stress-activated protein kinases, which activate the inhibitory response to protein synthesis [50]. The JNK protein is encoded by three genes, i.e., MAPK8 (JNK1), MAPK9 (JNK2), and MAPK10 (JNK3), and alternately spliced to produce ≥ 10 isomers. JNK1 and JNK2 are expressed in almost every cell, while JNK3 is mainly expressed in the brain [51]. The carcinogenic function of JNKs is related to their ability to phosphorylate Jun and activate AP1. In contrast, their antitumor effect may be related to the apoptotic activity [52]. In addition, JNK1 and JNK2 play different roles in cancer, promoting or inhibiting tumor formation [53]. There are also contradictions in functional research results of the JNK signal in OSCC. Table 1 summarizes studies on the JNK/MAPK signaling pathway associated with OSCC.

The study results of the JNK/MAPK signaling pathway in OSCC are controversial. Although some studies support the carcinogenic effect of JNK, others show that JNK plays a tumor-inhibitory role in HNSCC [63], which requires more studies to clarify its role in OSCC.

#### 2.3.3. p38/MAPK Signaling Pathway in OSCC

P38 kinase was originally screened and defined in drugs that inhibit the TNF-mediated inflammatory response [64]. The p38-MAPK pathway and some physiological changes in cells, such as growth signal transmission, the ability of unlimited replication, and apoptosis, angiogenesis, invasion, or metastasis prevention, are involved in transformation [65]. P38 is a conserved serine–threonine protein kinase, which can be activated by various extracellular inflammatory factors (e.g., TNF-α, IL-1), bacterial lipopolysaccharide, lipopolysaccharide, chemokine, and ultraviolet light. Activated p38 MAPK regulates cell function by regulating expression activities of downstream enzymes and transcription factors [66]. p38 MAPK activation is necessary for normal immune processes and inflammatory responses. It promotes key regulators of pro-inflammatory cytokine biosynthesis through transcription and translation, and thus, the components of this pathway become potential therapeutic targets for autoimmune and inflammatory diseases [67]. Simultaneously, it participates in tumorigenesis and ischemia-reperfusion injury [66]. In OSCC, p38 signal inhibition can reduce the tumor proliferation rate and reduce inflammation caused by the tumor [68,69]. Angiogenesis plays a key role in tumor progression, providing nutrition and oxygen for tumors and eliminating metabolic waste and carbon dioxide. Continuous neovascularization promotes tumor growth and diffusion [65,70]. P38α can control the growth of cancer cells and tumor-induced angiogenesis and lymphangiogenesis. It is a positive regulator in the tumor microenvironment of OSCC [23]. Banerjee et al. found that glycophorin receptor 2 (GALR2) induces angiogenesis by secreting pro-angiogenic cytokines mediated by the p38/MAPK signaling pathway, vascular endothelial growth factor (VEGF), and IL-6. In addition, GALR2 activates the small GTP protein Rap1b, which induces the inactivation of p38-mediated tristetraprolin (TTP). TTP is an RNA-BP that downregulates angiogenic factors, such as IL-6, VEGF, and IL-8, produced by tumor and inflammatory cells [71,72,73]. Its function is to destroy the stable transcription of cytokines, increase the secretion of pro-angiogenic cytokines, and promote angiogenesis in vitro and in vivo. In OSCC cells with GALR2 overexpression, p38 inhibition activates TTP and reduces cytokine secretion. TTP inactivation increases IL-6 and VEGF secretions [24]. IL-6 is a biomarker with low disease-specific survival [74], and VEGF elevation is associated with reduced recurrence time [75]. In HNSCC cases with low differentiation, p38 activation is more obvious and associated with a poor prognosis. The P38/MAPK signaling pathway is related to apoptosis and autophagy. Treating cells with the p38 MAPK (SB203580) or JNK1/2 (sp600125) inhibitor can promote/weaken G2/M phase arrest, apoptosis, and autophagy of cancer cells, respectively [76].

#### 2.3.4. MAPK Signaling Pathway and Immunity

Tumor cells downregulate immune cells in the tumor microenvironment to obtain tumor-promoting activity. MAPK is a central molecule of signal transduction regulating cell function. P38a MAPK participates in inflammation. It can produce proinflammatory cytokines [77], and acute inflammation can lead to cancer [78]. In the inflammatory microenvironment of OSCC, p38 αMAPK produces proinflammatory cytokines TNF-α, IL-1B, and IL-6, and plays a role in cancer progression [79]. Around MAPK-mutated HNSCC tumor cells, there exists a tumor microenvironment with high CD8+ T-cell inflammatory immunoreactivity, resulting in an increased endogenous lytic activity. These differences are evident in OSCC, suggesting that the ability of MAPK pathway mutation to predict disease in OSCC may be stronger than that of TMB. Pan pathway immune profiling studies revealed that MAPK-mutant tumors are the only “CD8+ T-cell inflammatory” tumors with an inherently hyperimmune responsive and structurally cytolytic tumor microenvironment. Immunoreactive MAPK-mutant models of HNSCC show massive in situ recruitment of cell-active or dead in situ CD8+ T cells. Consistent with the CD8+ T inflammatory phenotype, patients with MAPK-mutant OSCC had a 3.3–4.0-times longer survival time than patients with WT receiving anti-PD1/PD-L1 immunotherapy, independent of the tumor mutation burden. The pan-cancer prognosis of patients is consistent. MAPK mutations may recognize the high inflammatory/cytolytic activity of CD8+ T cells in patients with OSCC. p38 inhibitors have shown some success in treating and limiting adverse sequelae of inflammatory diseases and are a potential adjuvant therapy for OSCC [80].

IL-8 is closely related to the MAPK signaling pathway in OSCC. Leong et al. showed that it induces p-p38 MAPK and p-ERK expressions in HNSCC cells and downregulates p-JNK expression. It can increase NF-κB pathway expression in OSCC, suggesting that it may regulate MAPK and NF-κB pathways to regulate inflammatory response [22]. ROS or calcium ions activate p38MAPK, ERK [81], and JNK, which then affect the activation and transcriptional activity of hypoxia-inducible factor (HIF)-1. In addition, ERK leads to ser276p65/rela NF-κB phosphorylation of B, activating this transcription factor [82]. These processes lead to increased expression of some inflammatory genes and cell response to proinflammatory factors, particularly cyclooxygenase-2 [83], CC motif chemokine ligand 2/monocyte chemoattractant protein 1 (MCP-1) [84], CXC motif chemokine ligand (CXCL) 1/growth-related oncogene-α [85], CXCL8/IL-8 [86], and IL-6 [87]. They are inflammatory mediators involved in various tumor processes [88,89]. Figure 2 lists some well-studied key molecular pathways.

### 2.4. MAPK Signaling Pathway in the EMT Process in OSCC

Overcoming intercellular adhesion and invading surrounding tissues is the main feature of the transformation from benign lesions to metastatic cancer. EMT is key to this transformation. It is molecularly characterized by loss of E-calmodulin and increased expression of mesenchymal markers, including, for example, n-calmodulin, snail, fibronectin, and wave proteins [90,91]. The loss of E-cadherin is closely related to a poor prognosis [92]. As a result, the discovery of potential EMT blockers in patients with OSCC may reveal directions for novel therapies. Snails can regulate transcriptional inhibition of epithelial marker E-cadherin during EMT [93]. MAPKP38 increased significantly in tumor cell lines with high snail expression. The P38 interacting protein (p38ip) is a human analogue of the yeast spt20 protein. It is a subunit of histone spt3-taf9-gcn5 acetyltransferase. P38 binds to and stabilizes p38ip, resulting in enhanced transcription. P38-p38ip is involved in snail-induced downregulation of E-cadherin and cell invasion in OSCC. The transcription inhibitor snail plays a direct role in the downregulation of E-cadherin, while the proinflammatory mediator upregulates snail, thus affecting the cycle of inflammation-promoting tumor progression [94]. Cui et al. [95]. found that protein kinase D3 regulates PD-L1 expression and EMT in OSCC through ERK1/2. In OSCC, transient knockout of ERK1/2 can cancel PD-1/PD-L1-induced EMT. Similarly, ERK1/2 activation is affected by PD-L1 knockdown. PD-L1 interacts with PD-1 expressed by activated T cells, B cells, natural killer cells, some dendritic cells, and tumor-associated macrophages, thereby activating the PD-1/PD-L1 pathway. In contrast, activation of this pathway can inhibit the anti-tumor function of the same immune cells, decreasing anti-tumor immunity [96,97].

### 2.5. MAPK Signaling Pathway as a Therapeutic Target in OSCC

As a monotherapy, MAPK inhibitors are often ineffective and do not significantly affect the anti-proliferation of tumor cells, possibly by activating alternative EGFR downstream pathway targets to compensate for the inactivation of MAPK/JNK (i.e., adaptive rewiring) [68]. The latest molecular target-related studies related to the MAPK signaling pathway are summarized in Figure 3.

BRAF is an action target of targeted drugs. *RAF* family includes *BRAF*, *ARAF*, and *CRAF*. *BRAF* participates in the formation and development of malignant tumors. Before the wide application of genomic medicine, imprecise drug treatment of the MAPK pathway using MAPK pathway inhibitors failed in clinical trials. However, precise drug therapy has progressed. The V600E mutation using *BRAF* inhibitors alone has prolonged the survival of many patients with melanoma, thyroid cancer, and non-small cell lung cancer [98].

RAS is a GTP-binding protein, which mainly includes HRas/KRas/NRas, with 85% amino acid homology with each other. They are important targets for cancer research. KRas is the most common subtype in the Ras family, accounting for 85% of the total number of Ras gene mutations. KRas mutation was found in 90% of pancreatic ductal carcinoma cases. In addition, in melanoma, Ras mutation was found in 28% of cases, with NRas as the main mutant (93%) [99,100,101]. Although Ras is closely related to the occurrence and development of tumors, no targeted drugs in the market directly target Ras. The main reason is that the Ras protein has no characteristic, nearly spherical structure, and no obvious binding site. Therefore, it has been regarded as a “non-drug target” for a long time. However, covalent inhibitors and the emergence of targeted degradation agents have helped target KRas mutants [102,103]. Because Ras protein activation requires membrane localization, inhibition of Ras post-translational modified drugs, such as tipifarnib, can reduce MAPK pathway signal transduction and inhibit proliferation, apoptosis, and neovascularization [104].

MEK1/2 is a downstream protein of Ras and Raf; therefore, it can also be used as a target. Currently, the approved inhibitors of this target are mainly used in melanoma and lung adenocarcinoma, such as trametinib and selumetinib [105,106]. In recent years, more potent MEK inhibitors have been developed, with great prospects in treating OSCC and other cancers. For example, Trametinib is a promising OSCC MEK1/2 inhibitor. Of the 17 enrolled patients, nine (53%) had clinical to pathological tumor decline, suggesting that Trametinib significantly reduced Ras/MEK/ERK pathway activation and clinical and metabolic tumor response in patients with OSCC [107]. In a recent phase-II trial (nct02383927), the total effective rate of tritifanib in patients with HNSCC was 53% (13/23 patients) [108]. Somatic mutations of the mapk1 (ERK2) gene exist in approximately 5% of patients with HNSCC [109]. Allen V. et al. reported the first special responder with stage-III advanced oral cancer, which was treated with erlotinib for 13 days [110]. Whole-exome characteristics of pre-treatment biopsy showed the presence of somatic mapk1p E322k mutation, which can over-activate EGFR signal 61 by promoting the release of the autocrine amphoteric regulatory protein by OSCC cells and producing hypersensitivity to the EGFR kinase inhibitor erlotinib to over-activate EGFR [111]. In Hoi-Lam’s study, both MAPK1p.R135K and MAPK1p.D321N mutations activated EGFR and showed high sensitivity to erlotinib in vivo [109]. *MAPK* mutation regulates the activation of ERBB3 in OSCC, which is a new target of OSCC. Its activation level (p-erbb3) was significantly correlated with poor survival in patients with OSCC [21]. Currently, various ERBB3 inhibitors are under development. Among them, the anti-ERBB3 monoclonal antibody cdx-3379 showed antitumor activity, resulting in tumor shrinkage in 42% of patients with HNSCC [112].

CD8+ T cells are effective anti-tumor immune cells. They are significantly increased in OSCC with *MAPK* mutation, which may be related to the PD1 inhibitor response. A retrospective analysis of the anti-pd1 immunotherapy cohort showed that mutations in the MAPK pathway could predict the outcome of anti-pd1 immunotherapy in patients with advanced and metastatic oral cancer, and the median survival time was two to three times that of patients with WT [111,113,114]. These results suggested that the treatment of OSCC may require the exploration of effective MEK/MAPK inhibitors.

Despite advances in molecular diagnostics in recent years, cetuximab is currently the only targeted drug approved by the Food and Drug Administration [115]. Cetuximab, a monoclonal antibody against EGFR, has been the standard of care for OSCC for many years. However, the efficiency of cetuximab monotherapy is low (10–13%), and most patients gradually develop resistance even after initially producing a good therapeutic effect [116,117]. Therefore, OSCC remains a challenging disease to treat. We summarized the MAPK signaling pathway-related targeted therapeutic studies in recent years and summarized their specific targets of action (Table 2).

The MAPK pathway is also closely related to microRNAs (miRNAs) and long non-coding RNAs (lncRNAs) in OSCC. For example, BRAF-activated long non-coding RNA (BANCR) can promote the proliferation, migration, and invasion of OSCC cells through the MAPK pathway in OSCC, which may become a potential prognostic index and therapeutic target of OSCC [118]. In Jin et al.’s study [118], vitamin D inhibited the growth of OSCC cells through the lncrna lucat1-mapk signaling pathway. Mir-148a regulates the activity of the ERK/MAPK signaling pathway in OSCC through IGF-IR. IGF-IR can reverse the antitumor effect of mir-148a in OSCC [119]. Despite these encouraging results, the current research has many limitations, which should be resolved through future research.

### 2.6. Resistance of Cancer Cells to Drugs Targeting the MAPK/ERK Signaling Pathway

Although the drug therapy for OSCC has high efficacy and selectivity, OSCC is prone to drug resistance, in which the MAPK pathway plays an important role. Chemoresistance of tumor cells is mainly due to the lack of initiation and regulation of apoptosis. NRAS or MEK activation mutation, RAF amplification, RAF heterodimerization, BRAF alternative splicing, and NF1 deletion are the causes of acquired drug resistance [120]. In OSCC, most anticancer drugs induce apoptosis. Apoptosis can be initiated by mitochondria (endogenous pathway) or cell death receptors (exogenous pathway), activating the caspase cascade and apoptosis [122].

Inhibition of the p38 MAPK signaling pathway can reduce the tolerance of gastric cancer cells to Adriamycin treatment [123]. The p38 MAPK inhibitor combined with metformin improves cisplatin sensitivity in cisplatin-resistant ovarian cancer [124]. Inhibition of the MAPK/ERK signaling pathway reduces chemoresistance of small-cell lung cancer [125]. However, inhibition of MAPK signal transduction is insufficient to affect drug resistance of advanced prostate cancer cells, and the response to chemotherapeutic compounds does not depend on Raf/MEK/ERK signal transduction [126]. Resveratrol may lead to cell cycle arrest and subsequent apoptosis and autophagy to overcome drug resistance [127]. Chang et al. found that resveratrol decreased phosphorylations of ERK and p38, blocking their activation. While inhibiting the phosphorylation of ERK and p38, MMP2 and MMP9 expressions decreased significantly, and migration and invasion were significantly inhibited [128].

## 3. Perspectives and Conclusions

Despite research advancements, the prognosis of OSCC is unsatisfactory. Novel therapeutic strategies, target-specific drugs, and non-invasive, highly specific biomarkers improve the survival of patients with OSCC. However, questions and challenges persist.

First, although pro- and anti-cancer effects of the MAPK signaling pathway have been investigated in OSCC, the different expression levels of the MAPK signaling pathway should be elucidated in patients with OSCC with different clinical stages and corresponding regulatory programs, thus providing better clues for the early diagnosis and the treatment of tumor progression. Second, changes in MAPK signaling pathway-related markers were detected in peripheral blood mononuclear cells, particularly in patients treated with and after radiotherapy for OSCC. Therefore, in the future, protein markers of OSCC may be used as diagnostic indicators for clinicians for therapeutic interventions and as predictive markers by introducing proteomics technologies [80]. Third, because the MAPK signaling pathway is poorly conserved in patients at different stages, the molecularly targeted therapy has some challenges. Imprecise pharmacotherapy of the MAPK pathway using MAPK pathway inhibitors has failed in clinical trials with relatively large side effects [104]. Potential solutions to these problems have been mentioned in this paper, such as the exploration of Ras protein signatures and new binding sites that enable more drug-targeted therapies [102,103].

In recent years, a retrospective analysis of anti-PD1/PD-L1 immunotherapy cohorts has suggested that mutations in the MAPK pathway can predict the outcome of anti-pd1 immunotherapy in patients with advanced and metastatic oral cancer, enabling MEK/MAPK inhibitor-targeted therapy [110,114,115]. In addition, novel miRNA- and lncRNA-related studies, which suggest them as potential prognostic indicators and therapeutic targets for OSCC, have shown great potential in clinical applications [129].

In SCC, the catalytically active deubiquitinase Ubiquitin-specific peptidase 28 (USP28) is strongly expressed and stabilizes the essential squamous transcription factor ∆Np63 and c-JUN, and these changes increase the abundance of the proto-oncogene JUN [130]. The role of USP28 in squamous cell carcinoma has recently been elucidated, with its inhibition reducing the tumor growth rate and increasing cell death [131]. The effectiveness of USP28 inhibitors has been demonstrated in vivo [132]; however, little is known about USP28 in OSCC, and more studies are required to explore this role, which would be a promising target.

In summary, in this review, we summarized functional and potential molecular mechanisms of the MAPK signaling pathway in OSCC, molecularly targeted therapies, and interactions with the tumor microenvironment. In particular, we highlighted the potential of MAPK signaling pathway-related targets in future diagnostics and therapies, particularly emerging target therapeutic technologies, including Ras/MRK and immune checkpoint therapies targeting MAPK and the PD-1/PD-L1 axis. In conclusion, these results could provide a deeper understanding of the function of the MAPK signaling pathway in OSCC and present future prospects for this pathway in the diagnosis and treatment of OSCC.

## Figures and Tables

**Figure 1 cancers-14-04625-f001:**
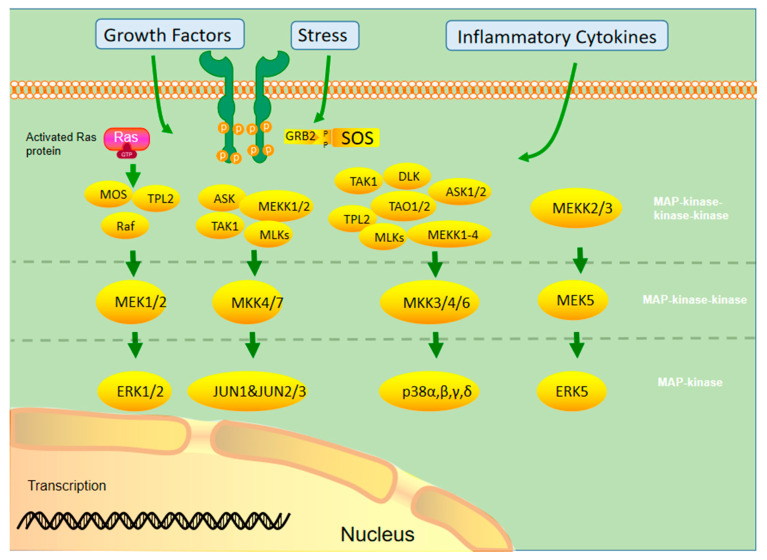
Different MAPKs associate with specific MAPK kinase (MAPKK) and MAPK kinase kinase (MAPKKK) to form a conserved three-stage enzymatic cascade (MAPKKK→MAPKK→MAPK), through which upstream signals are transmitted from MAPK to downstream nuclear transcription factors and cytoskeletal proteins to form a complete MAPK signaling pathway, which finally completes the regulation of cellular physiological activities.

**Figure 2 cancers-14-04625-f002:**
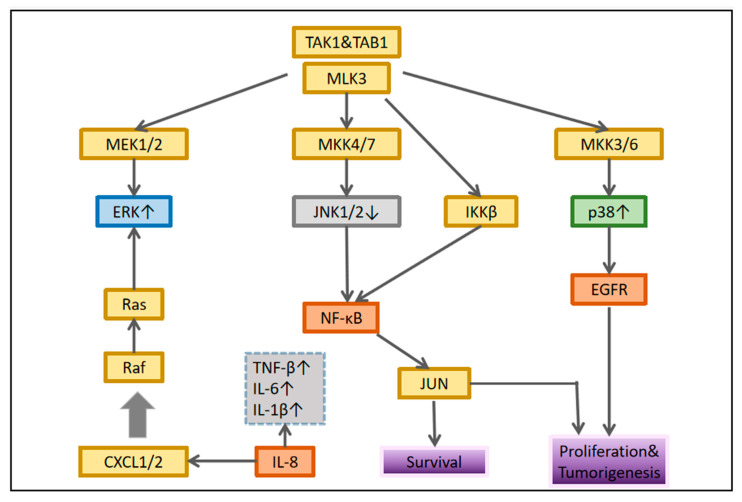
Interaction between MAPK and nuclear factor-κB signaling. In each cellular system, different connections are established that determine the biological response of the tumor.

**Figure 3 cancers-14-04625-f003:**
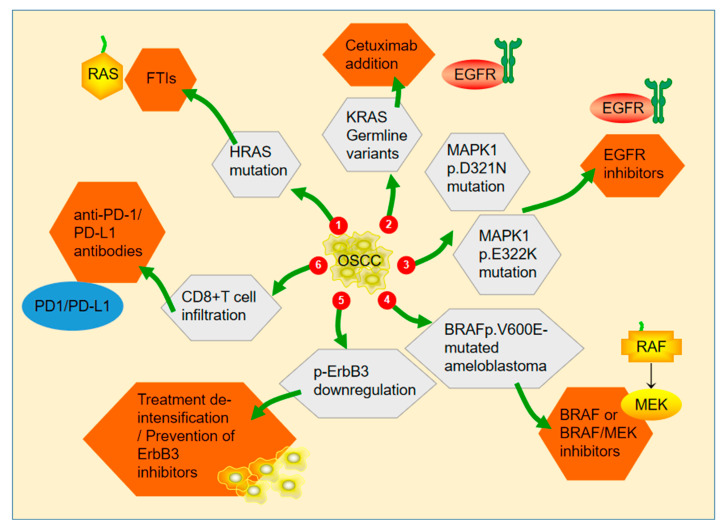
(1) Simultaneous inhibition of ERK and farnesyltransferase inhibits the growth of HRAS-mutated head and neck squamous cell carcinoma, and tipifarnib is currently being rapidly designated by the FDA for the treatment of HNSCC with HRA mutations. (2) Cetuximab is effective for the treatment of KRAS mutant types of HNSCC. (3) MAPK1 mutations can be targeted by EGFR inhibitors. (4) Glaucocytoma with BRAF p.V600E mutation is more sensitive to BRAF monotherapy or BRAF/MEK combination therapy. (5) Combined blockade of EGFR and ERBB3 promoted rapid tumor regression. (6) CD8+ T cell infiltration in MAPK mutant HNSCC may be an indicator of anti-PD-1 /PD-L1 inhibitor therapy.

**Table 1 cancers-14-04625-t001:** Advances in the JNK/MAPK signaling pathway associated with OSCC.

MAPK	Effects	Process	Site	Reference
JNK1/2	Promotes development	Negative cross-talk with the carcinogenic STAT3 signaling pathway. Chemical inhibition or selective targeting (via siRNA) downregulated STAT3 serine phosphorylation, accompanied by a modest increase in p-tyrstat3 levels. JNK activation could downregulate cell proliferation and viability and reduce cyclin D1 expression levels.	STAT3	[54]
ERK1/2, JNK1/2, p38	Promotes apoptosis	Protein G induces OSCC apoptosis by activating Akt, ERK1/2, p38, and JNK1/2, and JNK1/2 activation is associated with autophagy in tumor cells.	Protein G	[55]
JNK	Promotes development	Abnormal ubiquitination affects the corresponding JNK-dependent signaling pathway through the autophagy regulation mechanism.	Abnormal ubiquitination	[56]
JNK	Promotes apoptosis	C-Jun mediates Nur77 in the orphan nuclear receptor superfamily of glioblastoma multiforme, which plays a key role in ahpn/cd437-induced apoptosis. PC drugs promote Nur77 transfer from nucleus to cytoplasm in OSCC and induce cell apoptosis. Other apoptotic stimuli that induce Nur77 nuclear output, including TPA, VP16, and cisplatin, can activate JNK.	Nur77	[57]
JNK	Promotes apoptosis	JNK is involved in activating Bax, a pro-apoptotic Bcl-2 protein, after sunitinib treatment.	Bcl-2 protein	[58]
JNK	Promotes apoptosis	ROS production also mediates docetaxel-induced apoptosis of OSCC cells. ROS activates upstream kinase ASK1 of JNK. ASK1 activation must be tightly regulated according to the intensity and duration of stress (ROS), and various post-translational modifications, such as ubiquitination and methylation, participate in this tight regulation of activity and phosphorylation.	ROS, ASK	[59,60,61]
JNK	Promotes apoptosis	JNK activation is also involved in activating caspase induced in OSCC. It can activate caspase and reduce necrosis, apoptosis, cell cycle, and mito_x001e_chondrial membrane potential (∆Ψm), thus inducing OSCC cell apoptosis.	Caspase	[62]

**Table 2 cancers-14-04625-t002:** Oral squamous cell carcinoma-related targeted therapeutic molecules and targets.

Drug	Targets	Functions	References
Coronarin D	JNK	Induce JNK phosphorylation and promote apoptosis	[93]
Dehydrocrenatidine	ERK, JNK	Activation of ERK and JNK induces apoptosis.	[92]
Polyphyllin G	ERK, JNK, p38	Activation of ERK, Akt, p38, and JNK induces apoptosis of oral cells.	[87]
Xanthorrhizol	JNK, p38	Caspase-independent apoptosis was induced by ROS-mediated activation of p38MAPK and JNK.	[89]
PCH4	JNK	Induce JNK phosphorylation and promote apoptosis	[90]
Paclitaxel	JNK	Induce JNK phosphorylation and promote apoptosis	[91]
Demethoxycurcumin	JNK, p38	Induce JNK and p38 phosphorylation and promote apoptosis	[120]
Cetuximab	p38	Activation of p38 promotes skin toxicity.	[121]
Tipifarnib	ERK	Induce ERK phosphorylation and promote apoptosis	[117]

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
