# Peer review of "MAPK Signaling Pathway in Oral Squamous Cell Carcinoma: Biological Function and Targeted Therapy"

_cancers, 2022, doi:10.3390/cancers14194625_

Round 1

Reviewer 1 Report

Yuxi Cheng and colleagues summarize the role of MAPK signaling pathway in Oral Squamous Cell Carcinoma. The review is interesting and strictly focused on MAPK signaling pathway. However, I believe the review need to add recent literature studying the consequences of MAPK signaling activation and the possibility to target MAPK downstream factors in SCC tumors.

It has been reported that USP28 is an important downstream factor of MAPK signaling upon upregulation of MYC/JUN. In addition, it has been proved that USP28 inhibitor is an extremely promising therapy to treat SCC tumors. Please discuss the recent finders in a specific section. Particularly important it is to explain the important role of MYC in SCC tumorigenesis upon MAPK signaling activation and the potential of USP28 inhibitors to treat SCC tumors.

Upon include some important topics, the review will perfectly fit in Cancers.

It seems essential to include in the review the next points (Major points):

-       To explain the Activation of c-MYC upon MAPK signaling activation and the role of c-MYC in oral SCC tumorigenesis. Please check: https://pubmed.ncbi.nlm.nih.gov/35890188/

-       Discuss the activation of USP28 upon MAPK signaling activation and its important role in SCC tumors. It is essential to cite and discuss the next papers:

o   Paper discussing USP28 relevance and potential as therapy in SCC tumors: https://www.mdpi.com/2073-4409/10/10/2652

o   Papers demonstrating the activation of USP28 upon MAPK signaling activation via MYC/JUN and a potential therapy combining MAPK inhibitors to USP28 inhibitors: https://febs.onlinelibrary.wiley.com/doi/full/10.1002/1878-0261.13217

o   Papers demonstrating the important role of USP28 in SCC development and the potential of USP28 inhibitor to treat SCC (whole paper) and HNSCC (Appendix figure 3 using DET.562) : https://www.embopress.org/doi/full/10.15252/emmm.201911101

o   Paper demonstrating the relevant role of USP28 in SCC chemotherapy resistance and the potential combination of Cisplatin/USP28 inhibitors to treat SCC tumors: https://www.nature.com/articles/s41418-021-00875-z

Minor comments:

o   The paper intensively explains the potential of immunotherapy to treat SCC tumors but I believe it is important to expand the section related to DNA damaged therapies. Chemotherapy and radiotherapy are the main therapies to treat SCC tumors in combination to surgery and I believe it is important to explain in details and the current state of art of DNA damaged therapies in SCC tumors.

o   Could you explain the differences between C-RAF and A/B RAF recently reported (https://pubmed.ncbi.nlm.nih.gov/29395869/) and its potential role in the therapy of dysregulated MAPK tumors.

Author Response

Dear Sir,Madame, or Other:

Thank you very much for your important suggestions, we have read the literature you provided and learned a lot of new knowledge about both MYC and USP28.

We have augmented the article to include relevant content on MYC (lines 139-145) and USP28 (lines 460-467) and to provide an outlook on the potential role of USP28 in the treatment of OSCC.

Although USP28 has numerous intersections with MAPK-related molecular pathways in OSCC, no basic studies on USP28 in OSCC have been found. The excellent performance of USP28 in cancers such as lung adenocarcinoma is well established, and our group expects to focus more on the impact of its biological behavior in OSCC in future studies.

We are very sorry that we were not able to summarize the role of USP28 in OSCC well, this is to better make the content of the article relevant to the topic-OSCC. Thank you again for your valuable suggestions on our article, and we are more than willing to revise it again if there are more suggestions in the future.

Answer to question " The paper intensively explains the potential of immunotherapy to treat SCC tumors but I believe it is important to expand the section related to DNA damaged therapies. Chemotherapy and radiotherapy are the main therapies to treat SCC tumors in combination to surgery and I believe it is important to explain in details and the current state of art of DNA damaged therapies in SCC tumors.".

 Although the hypothesis that certain cancer genes are susceptible to these factors has been proposed and extensively studied in the past decades. Since then, chemotherapeutic agents and radiation therapy have been found to be effective in treating various cancers by inducing DNA damage. Many agents have been developed and new technical strategies have been explored in the war against cancer. However, there are still many challenges and unresolved issues that require further research, such as:

(1) Detailed molecular mechanisms of tumor cell DNA response to chemotherapeutic agents and radiotherapy;

(2) How cancer cells become resistant to chemotherapeutic agents and radiation therapy;

(3) Possible new and promising biomarkers to be investigated as new inhibitors or therapeutic agents;

(4) most importantly, the underlying biological mechanisms of DDR. With this information, effective cancer therapies can target DDR and ultimately prevent or cure cancer.

We are also very interested in targeting DNA damage therapies in OSCC, but unfortunately the length of this topic makes it difficult to provide an exhaustive account in this review, and perhaps we will summarize it separately or conduct new scientific studies in the future.

 Answer to question " Could you explain the differences between C-RAF and A/B RAF recently reported (https://pubmed.ncbi.nlm.nih.gov/29395869/) and its potential role in the therapy of dysregulated MAPK tumors.".

RAF protein kinases are downstream proteins of RAS in the MAPK pathway and can be activated by RAS through translocation, dimerization, and phosphorylation to participate in cell proliferation, survival, migration, and differentiation.RAF proteins include ARAF, BRAF, and CRAF/RAF1, of which BRAF gene mutations occur in 7% to 10% of cancers, while ARAF and RAF1 gene mutations are rarer . Some activated BRAF gene mutations cannot directly phosphorylate MEK, and BRAF forms a dimer with RAF1 to phosphorylate the MEK activation pathway.

BRAF inhibitors have shown good clinical benefit in the use of melanoma and colorectal cancer, but there are still no approved drugs for patients with BRAF non-V600E mutated tumors. Third-generation RAF inhibitors are in clinical trials, and such drugs are predicted to inhibit RAF dimers and be more effective in a wide range of RAF-dependent tumors. Examples include PLX-8394, BGB-283 (Lifirafenib), CEP-32496, etc. Ponatinib is a potent BRAF inhibitor that inhibits BRAF V600E monomer- and dimer-dependent melanoma cells, as well as the less potent mutant RAS-activated BRAF WT dimer. The ongoing NCI-MATCH study evaluates the efficacy of MEK inhibitors for BRAF non-V600E bearing solid tumors.

ERK inhibitors are currently in active development for patients resistant to BRAF inhibitors alone or in combination. reactivation of ERK kinase is one of the important mechanisms of BRAF resistance and ERK inhibitors have the potential to avoid or overcome resistance. The drugs in clinical trials include Uritinib (BioMed Valley Discoveries/Vertex), LY3214996 (Eli Lilly), LTT462 (Novartis), and JSI-1187 (JSI-Inda).

RX208, HL-085, a highly active BRAF V600E mutation selective inhibitor, can effectively block the RAF/MEK/ERK signaling pathway in BRAF V600E mutated cells, thus inhibiting the growth of BRAF V600E mutated tumors. HL-085 is a novel MEK inhibitor developed independently in China. Preclinical studies have shown that HL-085 combined with vemifenib is effective in BRAF V600 mutated solid tumors with potential anti-tumor activity. These two inhibitors targeting BRAF V600 mutation are in clinical trials in China.

There are few studies on BRAF-related drugs in OSCC, and we are equally looking forward to the emergence of better drugs for patients in the future.

                                                                                    Best Rrgards

Reviewer 2 Report

Thank you very much for this interesting and extensive and well written review on MAPK signalling. We just hope that the possible clinical applications will be seen in the context of the developments in other biomarkers.

Author Response

Dear Sir,Madame, or Other:

Thank you very much for your recognition of our article and valuable suggestions. Our research group has been committed to this aspect of research. The diagnosis and treatment of OSCC are in urgent need of more scholars to explore.

We will continue further research in this field in the subsequent research, hoping to improve the prognosis of patients through molecular targeted drug therapy. We are very grateful for your recognition.

Finally, wish you all the best in your research!

                                                                                          Best Rrgards

Reviewer 3 Report

The manuscript by Yuxi Cheng et al is a comprehensive review of MAPK pathway activation in oral squamous cell carcinoma encompassing the recent advancements in literature. Overall, the manuscript is a valuable addition with an updated overview of targeted therapeutic advances.

I have the following comment:

Lines 10-13:  The authors state “This review shares a summary of signaling pathways that have been studied by our group for a long time in cancer types, are particularly important in the development of oral cancer, and have never been reported in the literature before.” This sentence needs modification as it is somewhat misleading and appears to imply that MAPK pathways activation have never been reported in oral cancers.

Author Response

Dear Sir,Madame, or Other:

 Thank you very much for your approval and valuable corrections to our article. We strongly agree with your corrections and have revised the article based on them by changing lines 10-13 of the original article  “This review shares a summary of signaling pathways that have been studied by our group for a long time in cancer types, are particularly important in the development of oral cancer, and have never been reported in the literature before.”  In the newly revised article, lines 11-16, read "The aim of the present review was to summarize our studies on the signaling pathways in cancer types that are involved in the development of oral cancer and several mitogen-activated protein kinase-related molecular targeting technologies combined with immune checkpoint therapies to provide new therapeutic strategies for oral squamous cell carcinoma. In addition, we have provided a reasonable outlook, with a systematic basis, for future diagnosis and accurate treatment."

We will also continue to conduct follow-up studies on this topic in the hope of improving the prognosis of HNSCC patients through molecularly targeted therapies and other clinical applications.

Thank you again for your hard work, and finally, good luck with your research!

                                                                                            Best Rrgards

Round 2

Reviewer 1 Report

Thank you very much for your response. I fully recommend the publication of this review in the journal.

congrats for nice work!